# Marine-Derived N-Terminal Mitochondrial-Targeting Sequences Exhibit Antimicrobial and Anticancer Activities

**DOI:** 10.3390/ijms26178546

**Published:** 2025-09-03

**Authors:** Sun-Mee Hong, Kyu-Shik Lee, Kyuho Jeong, Jongwan Kim, Eun-Young Yun, Tae Won Goo

**Affiliations:** 1Department of Technology Development, Marine Industry Research Institute for East Sea Rim, Uljin 36315, Republic of Korea; hongsunmee@mire.re.kr; 2Department of Pharmacology, College of Medicine, Dongguk University, Gyeongju 38766, Republic of Korea; there1@dongguk.ac.kr; 3Department of Biochemistry, College of Medicine, Dongguk University, Gyeongju 38766, Republic of Korea; khjeong@dongguk.ac.kr; 4Department of Anatomy, College of Medicine, Dongguk University, Gyeongju 38766, Republic of Korea; dahyun@dongguk.ac.kr; 5Department of Integrative Bio-Industrial Engineering, Sejong University, Seoul 05006, Republic of Korea; yuney@sejong.ac.kr

**Keywords:** N-terminal mitochondrial-targeting sequences, marine organism, antimicrobial peptide, anticancer activity

## Abstract

The potential of N-terminal mitochondrial-targeting sequences (MTSs) as potent antimicrobial peptides (AMPs) has been previously reported. Building on this, 3923 mitochondrial proteins were identified from various marine organisms, among which 470 MTSs were predicted using MitoFates. Of these, 25 MTSs were synthesized and assessed for antimicrobial activity. All MTSs exhibited antifungal activity against *Candida albicans*, while 22 and 20 MTSs demonstrated activity against *Escherichia coli* and *Staphylococcus aureus*, respectively. Notably, the MTS of methylcrotonyl-CoA carboxylase subunit 1 (MCCC1-MTS) derived from swimming crab (*Portunus trituberculatus*) and the MTS of dihydrolipoamide branched-chain transacylase E2 (DBT-MTS) derived from herring (*Oncorhynchus keta*) showed strong antimicrobial activity against both Gram-positive and Gram-negative bacteria, as well as fungi. In addition, MCCC1-MTS markedly reduced the viability of multiple cancer cell lines with minimal cytotoxicity toward HaCaT cells and effectively suppressed the growth of A549-xenografted tumors in BALB/c nude mice without inducing weight loss. These findings demonstrate that MTSs derived from marine organisms function as potent AMPs with selective cytotoxicity toward cancer cells, further supporting previous evidence that protozoan MTSs represent novel AMP candidates.

## 1. Introduction

Cancer treatment has advanced with the development and implementation of various therapeutic strategies, including surgery, chemotherapy, radiotherapy, targeted therapy, and immunotherapy. Nevertheless, challenges such as low treatment efficacy in metastatic cancers, emergence of drug resistance, toxicity to normal tissues, limited applicability, and high treatment costs persist. Among these approaches, chemotherapy—despite being the most widely used—is associated with significant toxicity to normal cells, including immune cells, resulting in immunosuppression and heightened susceptibility to opportunistic infections. Sepsis is, in fact, a leading cause of death in individuals with cancer [1]. To reduce this risk, combination therapies involving chemotherapeutic agents and antibiotics have been employed. However, this strategy may contribute to resistance to both agents, ultimately compromising treatment efficacy. These limitations underscore the need for novel therapeutic approaches.

Antimicrobial peptides (AMPs) have attracted considerable interest due to their dual antimicrobial and anticancer activities. AMPs demonstrate a low propensity to induce resistance and display high selectivity for cancer cells, making them promising candidates for novel cancer therapies [2,3,4,5]. These peptides are essential components of the innate immune system, providing protection against microbial invasion, and have been identified across a wide range of organisms, from bacteria to mammals. Structurally, AMPs adopt diverse conformations—including α-helices, β-sheets, and extended structures—with many forming amphipathic α-helices carrying a net positive charge of +2 to +9 [6,7,8]. Their cationic regions interact with negatively charged bacterial membranes, while their hydrophobic domains insert into fatty acyl chains, ultimately disrupting membrane integrity or impairing membrane-associated functions [9].

To date, more than 3300 natural AMPs have been identified, exhibiting antimicrobial activity against a broad spectrum of pathogens, including bacteria, fungi, and viruses [7,10,11]. However, clinical translation of natural AMPs has been hindered by susceptibility to environmental factors such as pH, salt concentration, and proteases, as well as by hemolytic activity, low aqueous solubility, and high production costs. Consequently, no natural AMP has yet been approved for therapeutic use. To overcome these limitations, extensive efforts have been made to enhance AMP stability and activity through amino acid substitution and sequence modification [12,13]. Many AMPs in clinical trials are synthetic or chemically modified variants incorporating D-amino acids, N- or C-terminal chemical modifications, lipidation-mediated hydrophobic tuning, or cyclization strategies [13,14,15,16,17]. Nonetheless, large-scale production of synthetic and modified AMPs remains challenging due to high costs and potential metabolic side effects.

Mitochondria are membrane-bound organelles responsible for ATP synthesis in eukaryotic cells and are thought to have originated from ancestral prokaryotes through endosymbiosis [18]. Over evolutionary time, mitochondria retained only a small portion of their original genome, with the majority of mitochondrial genes transferred to the nuclear genome [19]. As a result, most mitochondrial proteins are synthesized in the cytoplasm and subsequently imported into the mitochondrial matrix, intermembrane space, or membranes. This import is directed by N-terminal mitochondrial targeting sequences (MTSs), which typically adopt cationic, amphipathic α-helical structures [20,21,22]. Unlike general AMPs, which often disrupt membranes nonspecifically, MTSs have evolved to selectively target mitochondrial membranes without affecting other cellular membranes. This selectivity is largely attributed to differences in the lipid composition, membrane potential, and associated protein import machineries across various intracellular membranes [23,24]. For example, the mitochondrial inner membrane exhibits a highly negative membrane potential and contains unique phospholipids such as cardiolipin, distinguishing it from the neutral and predominantly zwitterionic composition of the plasma membrane [23]. These features facilitate the specific interaction of MTSs with mitochondria. Such specificity suggests that MTSs function in a manner similar to cell-penetrating peptides, but with organelle-level selectivity.

Given the structural and electrostatic properties of MTSs and their evolutionary background, we hypothesized that MTSs may function as novel AMP candidates. In our previous study, we identified 26 MTSs from terrestrial organisms, over 50% of which exhibited antimicrobial activity. Furthermore, we demonstrated that the MTS of human mitochondrial transcription termination factor 2 suppressed pancreatic tumor growth in vivo [25]. This was the first study to highlight the therapeutic potential of MTSs as AMP-based agents. Building on these findings, the present study aimed to evaluate the antimicrobial and anticancer activities of MTSs derived from marine organisms to identify novel bioactive materials for therapeutic application.

## 2. Results

### 2.1. Screening of MTSs from Marine Organisms Exhibiting Antimicrobial Activity

A total of 3923 mitochondrial proteins were identified from 12 marine organisms by keyword-based searches and organism-specific filters using online bioinformatics databases, including NCBI and UniProt. Of these, 470 MTSs with a probability score ≥ 0.5 were selected using the bioinformatics tool MitoFates (Figure 1A) [26]. From this set, MTSs containing ≤ 30 amino acids and a net charge ≥ +2.0 were further analyzed using NetWheels and PEP-FOLD3 to assess helix-wheel and secondary structures [27,28]. Based on these analyses, 25 amphipathic α-helical MTSs were synthesized (Appendix A), and their antimicrobial activities were evaluated using a radial diffusion assay (RDA). All 25 synthesized MTSs exhibited antifungal activity against *C. albicans*; 22 showed antibacterial activity against the Gram-negative bacterium *E. coli*, and 20 were active against the Gram-positive bacterium *S. aureus* (Figure 1B). Notably, the MTS derived from methylcrotonyl-CoA carboxylase subunit 1 (MCCC1-MTS) exhibited antimicrobial activity against methicillin-resistant *S. aureus* (MRSA). Among these, MCCC1-MTS from swimming crab (*Portunus trituberculatus*) and dihydrolipoamide branched-chain transacylase E2 (DBT-MTS) from herring (*Oncorhynchus keta*) demonstrated the highest antimicrobial activity against all three pathogens (*E. coli*, *S. aureus*, and *C. albicans*) and were selected for minimal inhibitory concentration (MIC) analysis (Appendix A). The positive control, melittin, exhibited MIC values of 8 μg/mL (2.8 μM) against *C. albicans* and 16 μg/mL (5.6 μM) against *E. coli* and *S. aureus* (Figure 2). In comparison, MCCC1-MTS showed MIC values of 32 μg/mL (14.1 μM) against *E. coli* and *S. aureus*, and 64 μg/mL (28.2 μM) against *C. albicans* (Figure 2). DBT-MTS displayed lower antimicrobial activity, with an MIC of 128 μg/mL (49.3 μM) against all three pathogens.

### 2.2. Cytotoxicity Assessment of MCCC1-MTS and DBT-MTS Against Normal Cells

AMPs such as melittin possess potent antimicrobial activity and are considered promising candidates for treating infectious diseases [13,16,29,30,31,32]. However, their high cytotoxicity toward normal cells poses a major limitation for clinical application [33,34,35]. Therefore, the cytotoxicity of MCCC1-MTS and DBT-MTS was evaluated in normal cell lines. Melittin showed high cytotoxicity at concentrations ≥ 8 μg/mL in both Caco-2 and HaCaT cells (Figure 3). In contrast, MCCC1-MTS and DBT-MTS did not exhibit cytotoxicity up to 128 μg/mL (Figure 3). Furthermore, morphological assessment showed no visible cellular damage or structural disruption in cells treated with either MTS at concentrations up to 128 μg/mL. These findings indicate that the selected MTSs are promising AMP candidates with favorable biocompatibility and safety profiles.

### 2.3. Assessment of Physicochemical Stability of MCCC1-MTS and DBT-MTS

In addition to cytotoxicity, AMPs frequently face challenges related to instability under proteolytic and environmental conditions, including exposure to temperature, pH, and salinity, which limit their clinical applicability [36,37,38,39]. To address this, the physicochemical stability of MCCC1-MTS and DBT-MTS was evaluated using RDA. MCCC1-MTS demonstrated protease resistance comparable to melittin upon exposure to trypsin and chymotrypsin. In contrast, DBT-MTS completely lost its antimicrobial activity following trypsin treatment but retained strong resistance to chymotrypsin (Figure 4). Both MCCC1-MTS and DBT-MTS maintained high stability across a range of temperatures, pH values, and salinities (Figure 4). These findings suggest that MCCC1-MTS and DBT-MTS are more suitable for clinical applications via injection rather than oral administration.

### 2.4. Assessment of Cytotoxicity of MCCC1-MTS and DBT-MTS Against Cancer Cells

Cancer cells display negatively charged plasma membranes due to the translocation of phosphatidylserine from the inner to the outer leaflet via membrane flip-flop [40,41]. This electrostatic property facilitates specific interactions with AMPs, promoting cancer cell death [42,43,44]. In a previous study, we reported that hMTERF2-ts exhibits anticancer activity against multiple cancer cell lines, including MDA-MB-231 human breast cancer, HepG2 human hepatic cancer, HT-29 human colorectal cancer, and Panc-1 pancreatic cancer cells [25]. Based on this finding, flow cytometric analysis using annexin V (AV)/propidium iodide (PI) staining was conducted to investigate the anticancer effects and associated mechanisms of MCCC1-MTS and DBT-MTS. Both MTSs showed potent anticancer activity against A549 human lung cancer and MDA-MB-231 human breast cancer cells at concentrations above 100 μg/mL (Figure 5). In HT-29 human colorectal adenocarcinoma cells, over 50% reduction in viability was observed at 150 μg/mL for both peptides. In Panc-1 human pancreatic cancer cells, cytotoxicity was observed only with MCCC1-MTS at 150 μg/mL (Figure 5). MCCC1-MTS maintained high cell viability in HaCaT human keratinocytes at all tested concentrations, while DBT-MTS induced over 50% cell death at 150 μg/mL (Appendix A). The positive control, melittin, exhibited high cytotoxicity in both cancer and normal cells. These results indicate that MCCC1-MTS is a promising therapeutic candidate for lung and breast cancers, with minimal toxicity to normal cells.

### 2.5. Hemolysis Activity Analysis of MCCC1-MTS

To further evaluate the in vivo safety of MCCC1-MTS, its hemolytic activity was assessed following AV/PI staining results. The positive control, melittin, exhibited pronounced hemolytic activity at concentrations exceeding 8 μg/mL. In contrast, MCCC1-MTS showed negligible hemolytic activity even at 256 μg/mL (Figure 6). These findings support the potential of MCCC1-MTS as a highly biocompatible therapeutic agent.

### 2.6. Binding Analysis of MCCC1-MTS to Cancer Cell Membranes

As discussed previously, the anticancer activity of AMPs is mediated by their selective interaction with negatively charged cancer cell membranes [44]. To confirm that MCCC1-MTS targets the plasma membrane of A549 cells, fluorescence labeling experiments were performed. As shown in Figure 7, FITC-conjugated MCCC1-MTS localized to the plasma membrane of A549 cells after 1 h of treatment. After 5 h, FITC fluorescence was distributed throughout the entire cell. These results suggest that MCCC1-MTS binds to the A549 cell membrane, induces membrane disruption, and leads to cancer cell death.

### 2.7. Analysis of Pore Formation by MCCC1-MTS in Cancer Cell Membranes

Several AMPs induce cancer cell death by forming pores in the plasma membrane following binding [25,45,46]. To evaluate whether MCCC1-MTS induces pore formation in A549 cells, a trypan blue exclusion assay was conducted. The number of trypan blue-stained cells increased in a dose- and time-dependent manner following treatment with MCCC1-MTS (Figure 8). These findings indicate that MCCC1-MTS disrupts A549 cell membranes through pore formation, thereby inducing cancer cell death.

### 2.8. Evaluation of Anticancer Activity of MCCC1-MTS in an A549-Xenografted Animal Model

The anticancer activity of MCCC1-MTS against lung cancer was assessed using an A549 xenograft mouse model. Tumor growth was effectively suppressed in cisplatin-treated mice; however, these mice also exhibited significant body weight loss (Figure 9). In contrast, MCCC1-MTS administered at 20 mg/kg inhibited tumor growth without causing weight loss (Figure 9). Additionally, no reduction in organ weights was observed in MCCC1-MTS-treated mice (Figure 10A), and serum levels of hepatic damage markers, including aspartate aminotransferase (AST) and alanine aminotransferase (ALT), as well as renal damage markers such as blood urea nitrogen (BUN) and creatinine, remained unaltered (Figure 10B). These results demonstrate that MCCC1-MTS effectively suppresses tumor growth without inducing systemic toxicity.

## 3. Discussion

MTSs are signal peptides of mitochondrial proteins characterized by a positively charged α-helical structure, which is essential for the accurate translocation of cytosol-synthesized mitochondrial proteins into mitochondria [21,47]. The amphipathic nature and positive charge of MTSs are critical determinants of mitochondrial targeting [47]. Given the structural resemblance between MTSs and α-helical AMPs, we hypothesized that MTSs may exhibit AMP-like activity, based on protein structure–function relationships [25]. Notably, the positive charge of MTSs enables their specific targeting to mitochondria without disrupting normal cell membranes. We further speculated that this property might facilitate selective interactions with the negatively charged membranes of cancer cells, potentially leading to anticancer activity. Our previous investigation demonstrated that human-derived MTSs exhibit both antimicrobial and anticancer activities without cytotoxic activity against normal cells [25]. Additionally, in vivo experiments have confirmed that human MTSs effectively inhibit Panc-1 tumor growth without loss of body weight [25]. Considering the evolutionary origin of mitochondria from ancestral bacteria through endosymbiosis, we hypothesized that marine-derived MTSs may also possess antimicrobial and anticancer activities.

In this study, all 25 selected MTSs demonstrated antifungal activity against *C. albicans*. Among these, 22 displayed antibacterial activity against the Gram-negative bacterium *E. coli*, and 20 exhibited activity against the Gram-positive bacterium *S. aureus* (Figure 1 and Appendix A). These findings are consistent with our previous results and support the potential of MTSs as AMP candidates. They also suggest that expanding the search across diverse species may uncover additional MTS-derived AMPs.

Among the marine MTSs evaluated, MCCC1-MTS from swimming crabs and DBT-MTS from herring exhibited notable anticancer activity (Figure 9). MCCC1-MTS, in particular, did not induce significant cytotoxicity in normal cells or hemolysis in red blood cells (Figure 3, Figure 6 and Appendix A). In contrast, DBT-MTS reduced HaCaT cell viability by approximately 60% at 150 μg/mL (Appendix A), deviating from the expected mitochondrial specificity of MTSs. Further analysis of amino acid composition and charge distribution revealed that DBT-MTS has a markedly higher proportion of hydrophobic amino acids (50%)—excluding glycine—compared to MCCC1-MTS (37%), which may enhance membrane interaction and nonspecific cytotoxicity to HaCaT cells. Additionally, DBT-MTS contains lysine residue, contributing to a strong positive charge that may intensify its interaction with negatively charged components of normal cell membranes. These features suggest that the combination of high hydrophobicity and lysine content could underlie its off-target cytotoxicity. A more detailed structure–activity relationship analysis will be necessary to elucidate the mechanisms governing the selective cytotoxicity of MTSs.

AMPs represent promising therapeutic candidates for treating infections caused by antibiotic-resistant bacteria. However, their clinical development is hindered by instability to proteolytic degradation, thermal and pH fluctuations, and potential cytotoxicity in normal cells. Various chemical modifications—such as C-terminal amidation, D-amino acid substitution, acetylation, and cyclization—have been explored to improve AMP stability and bioavailability [48,49]. In fact, all AMPs currently approved by the FDA or undergoing clinical trials have undergone chemical modification. As shown in Figure 4, MCCC1-MTS demonstrated protease stability similar to melittin and maintained activity under thermal, pH, and salinity stress. Additionally, MCCC1-MTS exhibited significantly lower cytotoxicity toward normal cells (Figure 3 and Appendix A) and minimal hemolytic activity compared to melittin (Figure 6). Intravenous administration of MCCC1-MTS at 20 mg/kg in A549-xenografted mice resulted in effective tumor suppression without body weight loss (Figure 9). These results highlight MCCC1-MTS as a stable and effective therapeutic candidate for cancer and infectious diseases, potentially eliminating the need for chemical modification.

The outer membranes of normal cells predominantly consist of uncharged phospholipids, limiting their interaction with anticancer AMPs [50]. However, in cancer cells, phosphatidylserine, a negatively charged phospholipid typically found in the inner leaflet, is exposed on the outer leaflet due to membrane flip-flop, disrupting membrane asymmetry and facilitating AMP binding [51,52,53]. This binding leads to pore formation, a key mechanism underlying AMP-induced anticancer activity. Several anticancer AMPs interact with cancer cell membranes, form pores, and subsequently target nuclear or mitochondrial membranes to induce cell death [25,45,54,55,56]. Although MTSs facilitate protein translocation across mitochondrial membranes, they do not appear to disrupt mitochondrial integrity. Instead, they may function similarly to cell-penetrating peptides, which can traverse membranes without inducing permeabilization or cytotoxicity in healthy cells or organelles [57]. In this study, MCCC1-MTS selectively bound to cancer cell membranes and induced pore formation (Figure 8). MCCC1-MTS treatment led to cancer cell death and showed in vivo efficacy without inducing toxicity in normal tissues (Figure 10). Biomarker analysis revealed no significant changes in AST, ALT, BUN, or creatinine levels, confirming the absence of liver or kidney toxicity. These results suggest that MCCC1-MTS exerts anticancer activity primarily through selective interaction with negatively charged cancer cell membranes, thereby ensuring biocompatibility and in vivo safety.

Cell death may occur via apoptosis, necrosis, or autophagy. While apoptosis and autophagy preserve membrane integrity, necrosis is characterized by membrane disruption [58,59]. Trypan blue exclusion assays confirmed that MCCC1-MTS binds to A549 cell membranes and causes membrane damage (Figure 8). Moreover, AV/PI staining showed a significant increase in PI-stained and AV/PI double-stained cells (Figure 5), indicating membrane integrity loss and necrosis-mediated cell death.

Patients with cancer are frequently administered prophylactic antibiotics due to increased infection risk stemming from immunosuppression. Bacterial infections, especially sepsis, are a leading cause of cancer-related mortality. However, antibiotic treatment may interfere with chemotherapy efficacy and contribute to antibiotic and anticancer drug resistance. Therefore, developing therapeutic agents with both antimicrobial and anticancer properties is essential to improve cancer treatment outcomes and reduce resistance. This study demonstrated that MCCC1-MTS exhibits broad-spectrum antimicrobial activity against Gram-negative and Gram-positive bacteria and fungi, while also showing robust anticancer activity in both in vitro and in vivo models (Figure 2, Figure 5, Figure 9 and Appendix A). These findings suggest that MCCC1-MTS may overcome limitations associated with antibiotic co-administration and help mitigate the emergence of resistance.

In conclusion, this study provides compelling evidence that marine-derived MTSs represent multifunctional AMPs with both antimicrobial and anticancer properties. These findings support the potential application of MTSs as AMP-based therapeutics and build upon previous research demonstrating their promise as dual-function therapeutic agents.

## 4. Materials and Methods

### 4.1. Bacterial Cells

Antimicrobial activities of MTSs were evaluated by *Escherichia coli* (Gram-negative; KCCM 11234), *Staphylococcus aureus* (Gram-positive; KCCM 40881), and *Candida albicans* (KACC 11282). The bacterial cells were purchased from the Korean Collection for Type Culture (Wanju, Republic of Korea).

### 4.2. Animal Cancer Cells and Normal Cells Culture

A549 human lung cancer cells, MDA-MB-231 triple-negative human breast cancer cells, HT-29 human colorectal adenocarcinoma, Panc-1 human pancreatic cancer cell line, and Caco2 human epithelial cell line were purchased from the Korean Cell Line Bank (Seoul, Republic of Korea). HaCaT human keratinocytes were gifted by Professor Su-Kyung Kim (Department of Pharmacology, Keimyung University School of Medicine). The cells were routinely grown in Dulbecco’s Modified Eagle Medium supplemented with 10% fetal bovine serum and 1% antimycotic/antibiotic solution at 37 °C in an atmosphere containing 5% CO_2_.

### 4.3. Screening of Mitochondrial Proteins and MTSs, and Determination of the MTSs Secondary Structures

Mitochondrial proteins were screened in 12 species of marine organisms—including swimming crabs, herring, prawns, shrimp, oysters, salmon, whales, sharks, sea cucumbers, cuttlefish, squid, and brown algae in NCBI (https://www.ncbi.nlm.nih.gov/; accessed on 10 May 2022) and Uniprot (www.uniprot.org; accessed on 13 May 2022) databases. Further screening for MTSs was performed using MitoFates Ver. 1.2 (https://mitf.cbrc.pj.aist.go.jp/MitoFates/cgi-bin/top.cgi; accessed on 24 May 2022), an online bioinformatics tool for mitochondrial presequence prediction. Proteins with a presequence score higher than 0.5 were considered MTS candidates. Subsequently, 25 MTSs were selected based on the following criteria: (i) sequence length of less than 30 amino acids, (ii) net charge higher than +2, and (iii) predicted amphipathic α-helix structure as determined by helical wheel projection tool (http://lbqp.unb.br/NetWheels/; accessed on 24 May 2022) and protein structure prediction (https://mobyle.rpbs.univ-paris-diderot.fr/cgi-bin/portal.py#forms::PEP-FOLD3; accessed on 24 May 2022).

### 4.4. MTS Synthesis

The selected 25 amphipathic MTSs with purity exceeding 90% were synthesized by Anygen Co., Ltd. (Gwangju, Republic of Korea). The synthesized MTSs were dissolved by autoclaved distilled water and stored at temperatures below −20 °C until use.

### 4.5. Disc Diffusion Assay

The antimicrobial activities of 25 MTSs were evaluated using radial diffusion assay (RDA). The antibacterial activity of MTSs against Gram-negative bacteria was estimated using *E. coli*, whereas the activity against Gram-positive bacteria was tested using *S. aureus*. *C. albicans* was used to evaluate the antifungal activity. RDA was performed as follows: bacteria were cultured to a final concentration of 1 × 10^6^ cfu/mL and mixed with an underlay gel solution (9 mM sodium phosphate, 1 mM sodium citrate [pH 7.4], 1% low electro-osmotic agarose, and 0.03% TSB). The mixture was poured into a 100 mm square plate and allowed to solidify. Wells were created in a toe-solidified gel and various concentrations of melittin or MTSs were loaded into the wells. The plates were incubated at 37 °C for 3 h. Subsequently, an overlay gel solution (6% TSB, 1% low electroendosmosis agarose) was added, and incubation continued for 18 h at 37 °C. The antimicrobial activities of melittin and MTSs were determined by measuring the diameters of the clear inhibition zones formed around the wells. Melittin (Sigma-Aldrich, Merck KGaA, Darmstadt, Germany) was used as a reference.

### 4.6. Determination of MIC

To determine MIC of MCCC1-MTS and DBT-MTS, bacterial strains were cultured in liquid Mueller–Hinton broth at 37 °C with shaking at 200 rpm for 18 h. The cultures were then diluted to a final concentration of 1 × 10^4^ cfu/mL, and 90 μL aliquots were dispensed into wells of a sterile 96-well microplate. Subsequently, 10 μL of serially diluted MTSs solution was added to each well. The plates were incubated at 37 °C for an additional 18 h. After incubation, the optical density at 600 nm was measured using an Epoch ELISA Reader (Agilent Technologies, Santa Clara, CA, USA) to determine bacterial growth and establish the MIC value. Melittin was used as a reference AMP.

### 4.7. Cell Viability Assay

The cytotoxic activities of MCCC1-MTS derived from swimming crabs and DBT-MTS derived from herring against Caco2 and HaCaT cells were assessed using the 3-(4,5-dimethylthiazol-2yl)-2,5-diphenyltetrazolium bromide (MTT) assay. 1 × 10^4^ cells were seeded into each well in a 96-well plate and attached for 24 h at 37 °C. Cells were cultured in conditioned medium containing various concentrations of MCCC1-MTS, DBT-MTS, or melittin for 24 h. The medium was removed, and the cells were incubated for 4 h in the dark after adding 50 μL of MTT reagent (Sigma-Aldrich; Merck KGaA, Darmstadt, Germany) in an each well. Subsequently, 100 μL of dimethyl sulfoxide was added to each well to dissolve insoluble formazan. The absorbance of solubilized formazan was measured at 530 nm using an Epoch ELISA Reader.

### 4.8. Physical and Chemical Stability Analysis

We analyzed the stability of MCCC1-MTS and DBT-MTS against proteases, heat, pH level variations, and salt stress by comparing the diameters of the clear inhibition zones with those of melittin by RDA using *E. coli*. To determine protease stability, two MTSs and melittin were incubated with 50 nM and 100 nM of trypsin or chemotrypsin for 4 h at 37 °C and then the proteases were inactivated by heating at 95 °C for 10 min. The MTSs and melittin were heated at 95 °C for 24 h and cooled for 10 min at room temperature to analyze heat stability, and they were exposed to solutions of varying pH levels (2–11) to estimate pH stability. To evaluate stability against different salt types, MCCC1-MTS, DBT-MTS, and melittin were dissolved in a solution containing 150 mM NaCl, 1 mM MgCl_2_, or 2.5 mM CaCl_2_.

### 4.9. Flow Cytometric Analysis

To confirm the cytotoxic effects of MCCC1-MTS and DBT-MTS on cancer and HaCaT cells, and to investigate the mechanism of cell death, flow cytometric analysis was performed using the AV/PI Apoptosis Kit (Biovision, Milpitas, CA, USA). 5 × 10^5^/well cells were plated into 6-well plates and incubated at 37 °C with 5% CO_2_ for 24 h to allow adhesion. The cells were then treated with conditioned medium containing varying concentrations of MCCC1-MTS, DBT-MTS, or melittin for 24 h. The conditioned medium was collected into a 15 mL conical tube to retrieve the floating cells, while adherent cells were detached using trypsin. The floating cells in the collected medium were combined with the detached adherent cells and centrifuged for 2 min at 3000 rpm at 4 °C to collect the cells. The cell pellet was resuspended using 1x binding buffer containing AV and PI and incubated for 30 min at 37 °C. The cells were analyzed using a CytoFLEX Flow Cytometer (Beckman Coulter, Brea, CA, USA).

### 4.10. Hemolysis Assay

The physiological safety of MCCC1-MTS was assessed by hemolytic activity analysis using human red blood cells (RBCs). This study was approved by the Institutional Review Board of Dongguk University (No.: DGU IRB 20200007-01). RBCs were prepared from human blood obtained from the Korean Red Cross by centrifugation at 2000 rpm for 30 min at 10 °C, followed by pellet collection. RBC pellets were washed by resuspending in phosphate-buffered saline (PBS) and centrifugation at 2000 rpm for 30 min at 10 °C thrice. Subsequently, RBCs were resuspended in PBS and 50 μL/well of RBC solution was loaded into a 96-well plate. Then, the RBC solutions were mixed with various concentrations of MCCC1-MTS, melittin, or 1% Triton X-100 solution, and the mixtures were incubated at 37 °C for 1 h and centrifuged at 2000 rpm for 30 min at 10 °C. Finally, the absorbance of the supernatant was measured at 405 nm using an Epoch ELISA Reader. The hemolytic activities of MCCC1-MTS and melittin were expressed as % absorbance of the sample versus the absorbance of 1% Triton X-100-treated. All the absorbance values were normalized to those of PBS.

### 4.11. Analysis of Interaction Between MCCC1-MTS and Cell Membrane

To investigate whether the marine MTSs specifically interacts with cancer cell membranes, the interaction between MCCC1-MTS and A549 cells was analyzed. A549 cells were seeded onto a collagen-coated coverslip in 6-well plates and attached for 24 h. Then, the cells were treated with 150 μg/mL of FITC-conjugated MCCC1-MTS for 1, 3, or 5 h. The cells were washed with Dulbecco’s PBS (Biosesang, Seongnam, Republic of Korea) and fixed with 4% paraformaldehyde (Biosesang) for 20 min at 4 °C. The FITC-conjugated MCCC1-MTS bound to the cell membrane was observed using a fluorescence microscope (Carl Zeiss NTS Ltd., Oberkochen, Germany).

### 4.12. Trypan Blue Exclusion Assay

To determine whether MCCC-1 MTS creates pores in the cancer cell membrane, A549 human lung cancer cells were stained with trypan blue (Sigma-Aldrich). 5 × 10^5^/well were seeded on 6-well plates, attached for 24 h, and cultured for 1 or 3 h after the treatment of 50 or 100 μg/mL MCCC1-MTS. The cells were then detached by trypsinization and harvested by centrifugation at 3000 rpm for 2 min. The cell pellet was resuspended with 20 μL of PBS, and the cells were stained using 20 μL of 0.4% trypan blue solution. Subsequently, the cells were counted using a hemocytometer.

### 4.13. Animal Experiment

The effect of MCCC1-MTS on tumor growth was evaluated in vivo using an A549 xenograft nude mouse model. The experiment was approved by the Institutional Animal Care and Use Committee of Dongguk University (IACUC-2024-03) and conducted in accordance with the Experimental Animal Use guidelines. The four-week-old female Balb/c nu/nu mice were bought from Orient Bio (Sungnam, Republic of Korea) and maintained for two weeks at 25 ± 2 °C in 50 ± 5% relative humidity with 12 h dark/light cycles. Then, 5 × 10^6^ A546 human lung cancer cells mixed with Matrigel were injected subcutaneously into the right dorsal area. MCCC1-MTS (10 or 20 mg/kg) or cisplatin (5 mg/kg) was intravenously injected when the tumor volume reached approximately 400 mm^3^. MCCC1-MTS was administered in triplicate each week for two weeks, and cisplatin was injected twice in the first week. The body weights and tumor sizes of the mice were measured twice a week for five weeks. The mice were sacrificed at thirty-three days after the first injection of MCCC1-MTS or cisplatin. The tumor volume in each mouse was determined using the following formula: tumor volume (mm^3^) = length × width^2^ × 0.5. The lungs, liver, spleen, and kidneys were removed from the sacrificed mice and their weights were measured. Blood samples were also collected and serum levels of ALT, AST, BUN, and creatinine were analyzed by an outsourced Korea Non-Clinical Technology Solution Center (Seongnam, Republic of Korea).

### 4.14. Statisical Analysis

All experiments were independently performed in triplicate, and the results were presented as mean ± standard deviation. Each result was statistically analyzed using one-way analysis of variance, followed by Tukey’s honest significant difference test or Fisher’s least significant difference test using SPSS Ver. 20.0 software (SPSS, Inc., Chicago, IL, USA). Differences were considered statistically significant at *p* < 0.05.

## Figures and Tables

**Figure 1 ijms-26-08546-f001:**
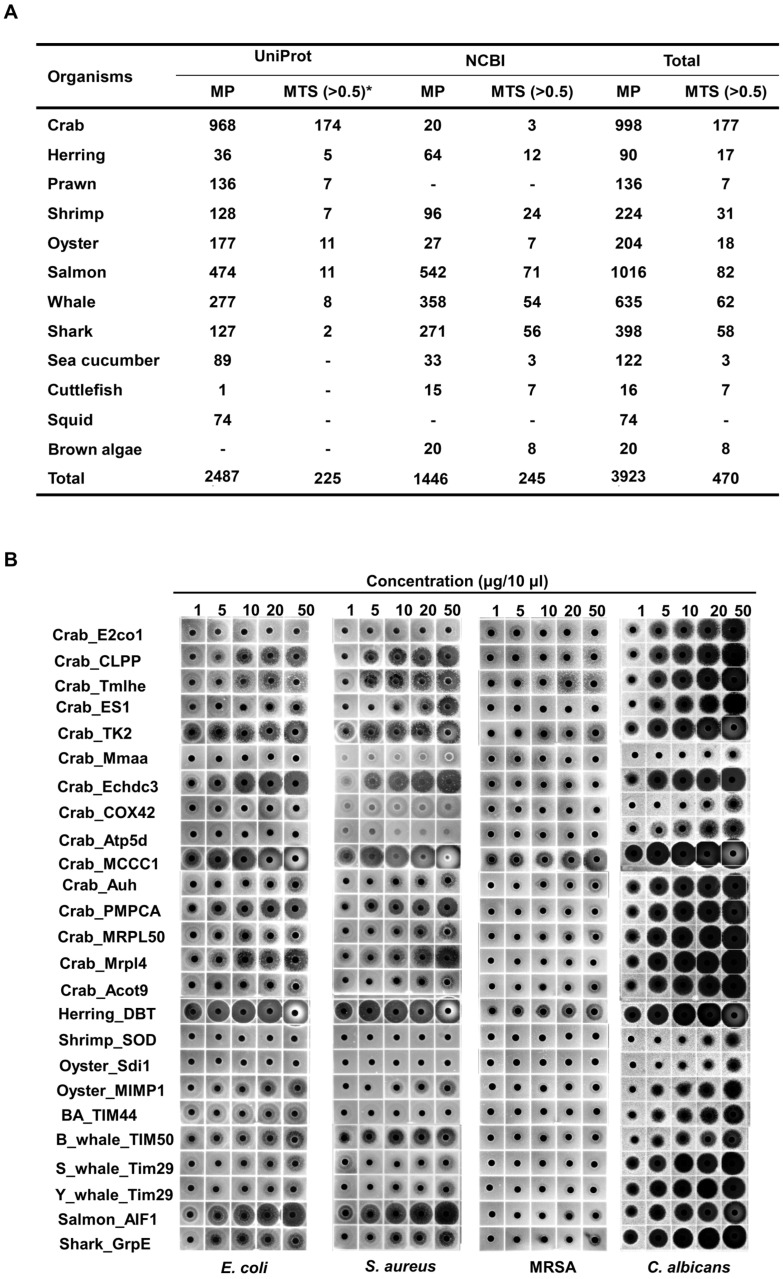
Screening of mitochondrial proteins and MTSs from marine organisms, followed by evaluation of antimicrobial activities of selected MTSs. (**A**) Mitochondrial proteins in marine organisms were screened using UniProt and NCBI, and their MTSs were identified using MitoFates (Ver. 1.2), a bioinformatic tool for evaluating possibility scores of MTSs. * indicates MTS probability score. (**B**) Selected MTSs were synthesized, and the antimicrobial activities against pathogens were identified using RDA.

**Figure 2 ijms-26-08546-f002:**
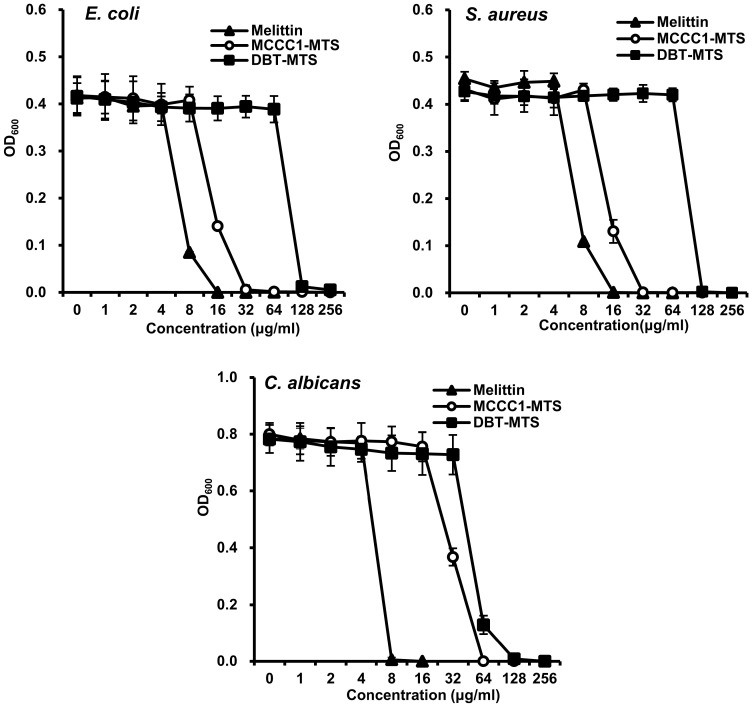
MIC of MCCC1-MTS and DBT-MTS against *E. coli*, *S. aureus*, and *C. albicans*. Each bacterial strain of 1 × 10^4^ cfu/mL was inoculated in liquid media containing various concentrations of melittin, MCCC1-MTS, or DBT-MTS, and cultured for 18 h. The optical density of the culture media was measured at 600 nm to determine the MIC of each AMP.

**Figure 3 ijms-26-08546-f003:**
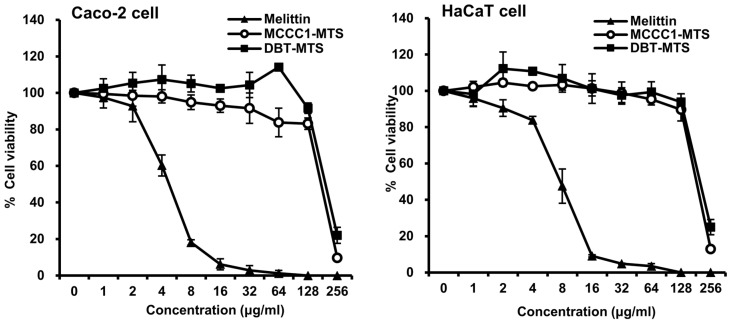
Cytotoxicity of MCCC1-MTS and DBT-MTS against Caco-2 and HaCaT cells. Cells were treated with various concentrations of melittin, MCCC1-MTS, or DBT-MTS for 24 h. Then, the cells were further incubated for 4 h in the dark after adding MTT reagent. After incubation, formed formazan crystal was dissolved. The absorbance of the solubilized formazan was measured at 530 nm to determine cell viability of each AMP.

**Figure 4 ijms-26-08546-f004:**
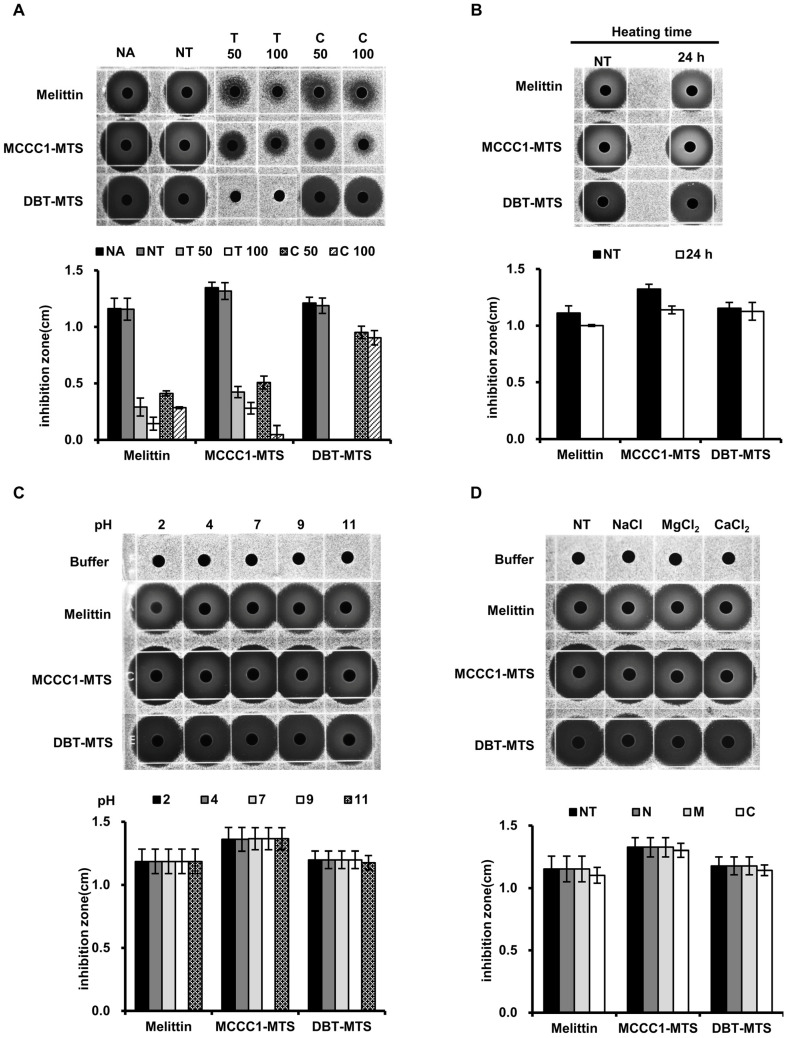
The analysis of MCCC1-MTS and DBT-MTS stability against proteases, heat, pH, and salts using RDA. (**A**) MTSs were treated by proteases. (**B**) MTSs were heated at 95 °C for 24 h. (**C**) MTSs were exposed to solutions of varying pH levels. (**D**) MTSs were dissolved in solutions containing 150 mM NaCl, 1 mM MgCl_2_, or 2.5 mM CaCl_2_. (**A**–**D**) Then, the MTSs were loaded onto each gel plate containing *E. coli* and were incubated for 18 h. Subsequently, the diameter of the clear inhibition zones formed on the gel plates were measured to determine antimicrobial activities of MTSs.

**Figure 5 ijms-26-08546-f005:**
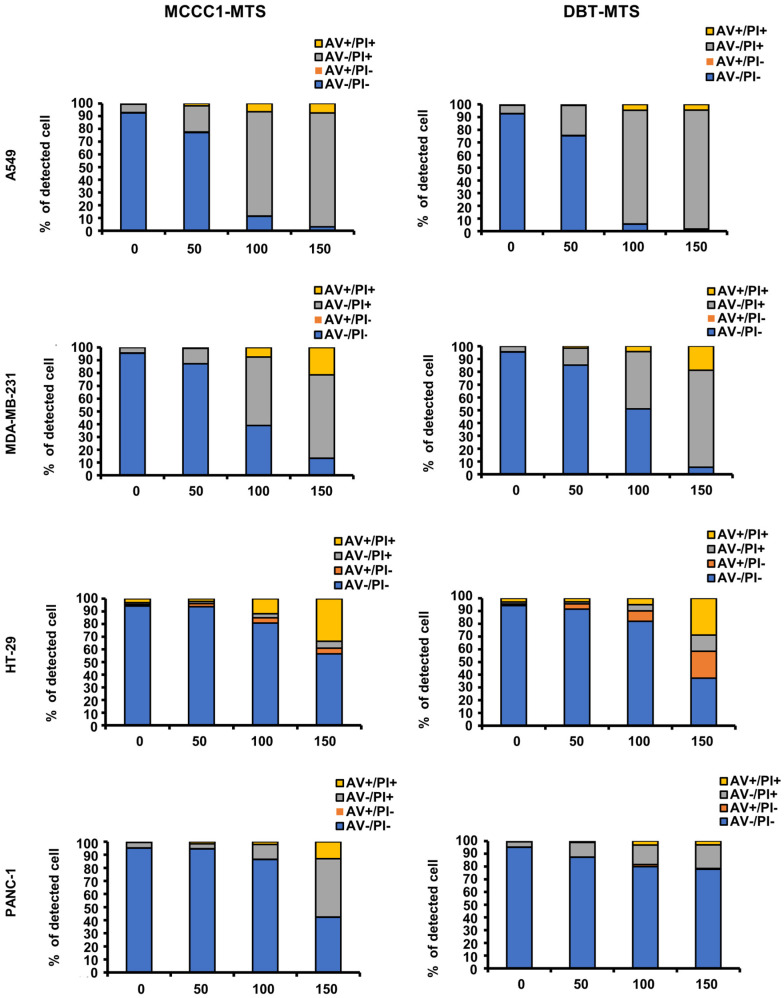
Flow cytometric analysis of MCCC1-MTS and DBT-MTS cytotoxic activity against various cancer cells. Cells were treated by various concentrations (0, 50, 100, and 150 μg/mL) of MCCC1-MTS or DBT-MTS for 24 h and stained by AV/PI.

**Figure 6 ijms-26-08546-f006:**
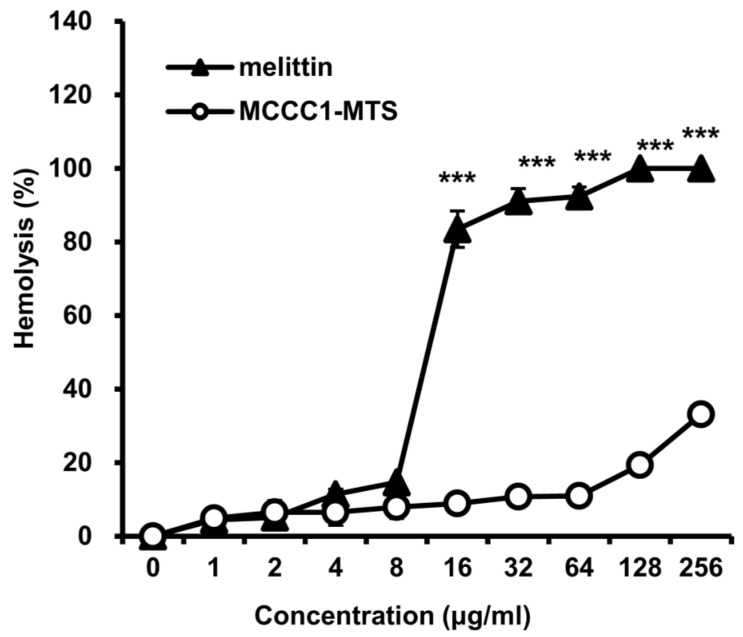
Analysis of hemolytic activities of melittin and MCCC1-MTS. Cells were treated by MCCC1-MTS or DBT-MTS for 24 h, then absorbance of RBC lysate was measured at 405 nm. Hemolysis (%) was determined by dividing the absorbance of RBC lysates treated with MCCC1-MTS or melittin by that of RBCs treated with 1% Triton X-100 and multiplying the result by 100. *** indicates *p* < 0.001 when compared to melittin.

**Figure 7 ijms-26-08546-f007:**
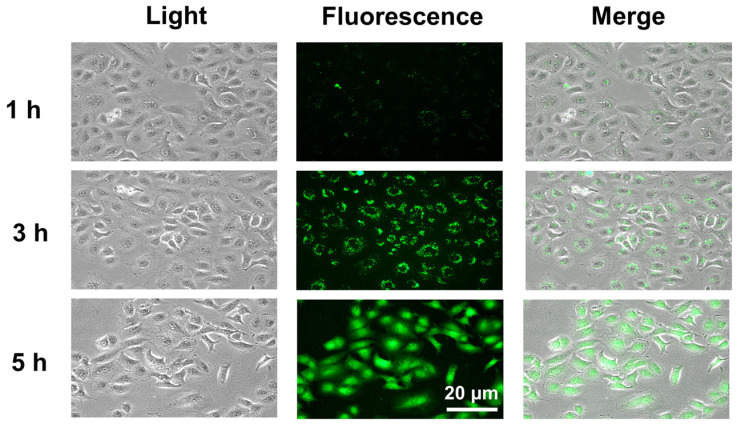
Evaluation of the interaction between MCCC1-MTS and A549 lung cancer cell membranes. Cells were treated with 150 μg/mL of FITC-conjugated MCCC1-MTS for 1, 3, or 5 h, and then FITC-conjugated MCCC1-MTS bound to cell membrane was observed.

**Figure 8 ijms-26-08546-f008:**
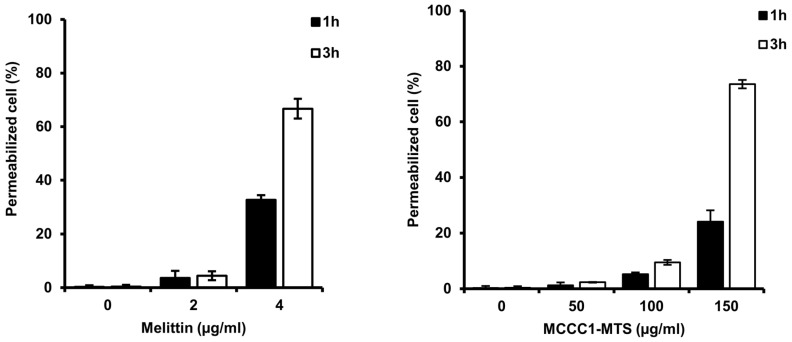
Analysis of cell permeabilization of MCCC1-MTS using trypan blue exclusion assay. Cells were treated with melittin or MCCC1-MTS for 1 or 3 h and stained using trypan blue. Then, stained cells were counted to identify permeabilized cells. Melittin was used as positive control.

**Figure 9 ijms-26-08546-f009:**
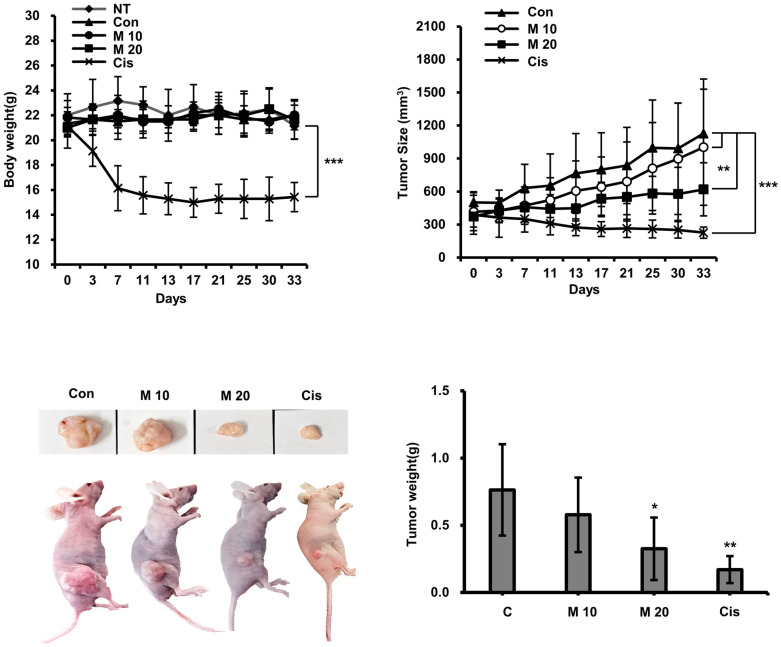
Analysis of anticancer activity of MCCC1-MTS. MCCC1-MTS or cisplatin was intravenously injected when tumor volume reached approximately 400 mm^3^. The body weights of mice and their tumor sizes were measured twice a week. Cisplatin was used as internal control. The analysis was performed by the first authors (Sun-Mee Hong and Kyu-Shik Lee), who was aware of group allocations during the allocation process, conduct of the experiment, outcome assessment, and data analysis. *, **, and *** indicate *p* < 0.05, *p* < 0.001, and *p* < 0.0001, respectively, compared to the control.

**Figure 10 ijms-26-08546-f010:**
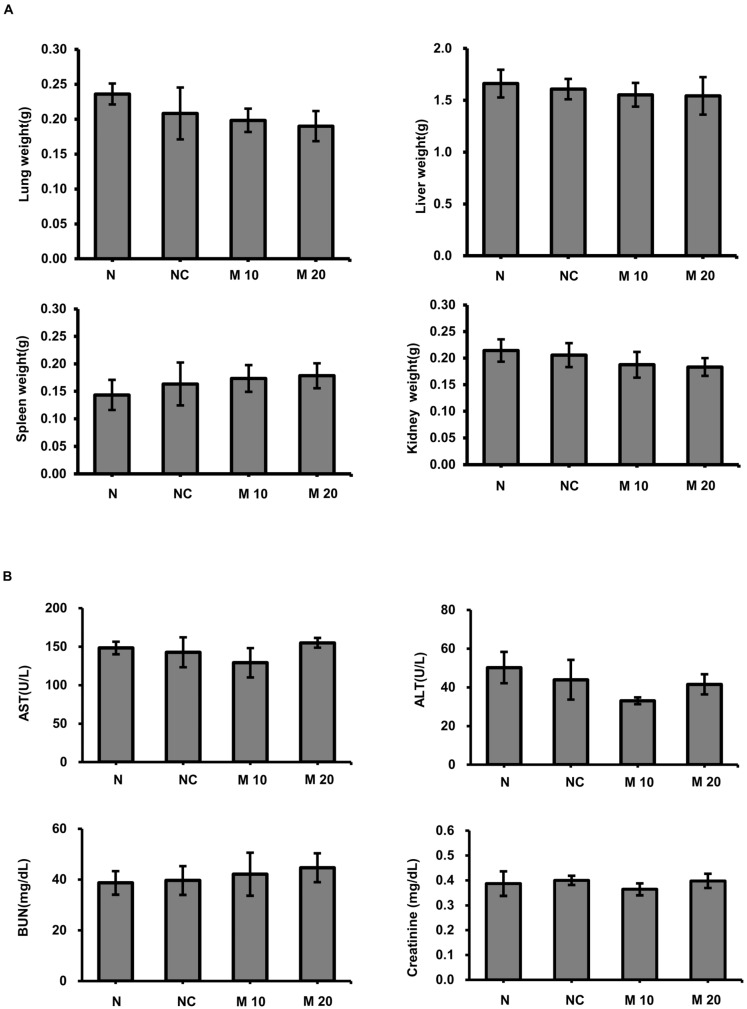
Evaluation of organ damage following MCCC1-MTS administration. (**A**) Weights of the lung, liver, spleen, and kidney were measured to assess potential tissue damage by MCCC-1 MTS injection. (**B**) Serum levels of AST, ALT, BUN, and creatinine were measured to evaluate liver and kidney function.

## Data Availability

The data presented in this study are contained within the article and Appendix A.

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
