# Peer review of "Marine-Derived N-Terminal Mitochondrial-Targeting Sequences Exhibit Antimicrobial and Anticancer Activities"

_ijms, 2025, doi:10.3390/ijms26178546_

Round 1
Reviewer 1 Report
Comments and Suggestions for Authors
- Abstract: I wouldn’t put the full HTTP address in the abstract (I would specify it in the main text instead), as that is likely to change in the future. For example, now I cannot access that web page, and I could not find the updated address.
- Introduction: As most AMPs are targeting membranes, I would explain more about how MTSs evolved to selectively target mitochondrial membranes without interacting with other cellular membranes. What is different among the different membranes (https://doi.org/10.1038/s41580-020-0210-7, https://doi.org/10.1016/j.biochi.2022.07.011).
- Results:
- “A total of 3,923 mitochondrial proteins were identified from 12 marine organisms using online bioinformatics databases, including NCBI and UniProt.” -> How did you identify them? Which tools did you use to explore those databases and fetch the needed data?
- “using NetWheels and PEP-FOLD3” -> Please add the references for those tools.
- Can you also show the activities in micromolar in the text? That will help with the comparison to Mellitin MIC values.
- Figure 5: The legend for the horizontal axis is missing for each panel (0, 50, 100, 150).
- Figure 7: “Fluorescenc” label is truncated in the image.
- Discussion: I wonder if those MTSs are affecting mitochondrial membrane permeability too. If they are naturally used to translocate molecules into the mitochondrial membrane, they should behave more as cell-penetrating peptides without damaging the mitochondrial membrane (https://doi.org/10.1016/bs.mie.2020.04.044). On the other hand, they look to disrupt other negatively charged cell membranes (such as bacterial and cancer ones). If they are not damaging mitochondrial membranes, maybe their anticancer mechanism of action is not based on apoptosis induction (pointing more to your suggested necrosis-mediated death).
- Materials and Methods:
- Bacterial cells: Please fix the italics formatting.
- “Screening of mitochondrial proteins and MTSs, and determination of the MTSs secondary”: if only NCBI and UniProt were used, that is fine, but the text should be adapted to drop “using online bioinformatics databases, SUCH AS” statements (there is a similar one in the Results section). Please also cite the relevant tools (i.e., Pep-fold, Netwheels, MitoFates…)
- “Determination of MIC” -> Which media was used?
Author Response
Thank you for your helpful and insightful comments. Now, we reply for your comments as below.
- Abstract: I wouldn’t put the full HTTP address in the abstract (I would specify it in the main text instead), as that is likely to change in the future. For example, now I cannot access that web page, and I could not find the updated address.
: Thank you for your kind and insightful comment. We fully agree with your comment. We erased the full HTTP address from the abstract.
- Introduction: As most AMPs are targeting membranes, I would explain more about how MTSs evolved to selectively target mitochondrial membranes without interacting with other cellular membranes. What is different among the different membranes (https://doi.org/10.1038/s41580-020-0210-7, https://doi.org/10.1016/j.biochi.2022.07.011).
: Thank you for your insightful comment. We have revised the introduction to include a detailed explanation of how MTSs selectively target mitochondrial membranes without interacting with other cellular membranes, incorporating the references you suggested.
- Results:
- “A total of 3,923 mitochondrial proteins were identified from 12 marine organisms using online bioinformatics databases, including NCBI and UniProt.” -> How did you identify them? Which tools did you use to explore those databases and fetch the needed data?
: Thank you for your insightful comment. We have revised the results section to clarify that mitochondrial proteins were identified by keyword-based searches and organism-specific filters using the NCBI and Uniprot databases.
- “using NetWheels and PEP-FOLD3” -> Please add the references for those tools.
: Thank you for your insightful comment. We have added the references for NetWheels and PEP-FOLD3
- Can you also show the activities in micromolar in the text? That will help with the comparison to Mellitin MIC values.
: Thank you for your insightful comment. We have shown the activities in micromolar in the text of section 2.1.
- Figure 5: The legend for the horizontal axis is missing for each panel (0, 50, 100, 150).
: Thank you for your insightful comment. We have revised the legend of Figure 5 by adding the missing horizontal axis labels (0, 50, 100, 150) for each panel.
- Figure 7: “Fluorescenc” label is truncated in the image.
: Thank you for your insightful comment. We have revised “Fluorescenc” label to “Fluorescence” in the image.
- Discussion: I wonder if those MTSs are affecting mitochondrial membrane permeability too. If they are naturally used to translocate molecules into the mitochondrial membrane, they should behave more as cell-penetrating peptides without damaging the mitochondrial membrane (https://doi.org/10.1016/bs.mie.2020.04.044). On the other hand, they look to disrupt other negatively charged cell membranes (such as bacterial and cancer ones). If they are not damaging mitochondrial membranes, maybe their anticancer mechanism of action is not based on apoptosis induction (pointing more to your suggested necrosis-mediated death).
: Thank you for your insightful comment. We have revised the Discussion section to explore the potential effects of MTSs on mitochondrial membrane permeability, incorporating the reference you suggested.
- Materials and Methods:
- Bacterial cells: Please fix the italics formatting.
: Thank you for your insightful comment. We have revised the format as the italics.
- “Screening of mitochondrial proteins and MTSs, and determination of the MTSs secondary”: if only NCBI and UniProt were used, that is fine, but the text should be adapted to drop “using online bioinformatics databases, SUCH AS” statements (there is a similar one in the Results section). Please also cite the relevant tools (i.e., Pep-fold, Netwheels, MitoFates…)
: Thank you for your helpful comment. We have revised the text to remove the pharse “using online bioinformatics databases, such as”. In addition, we have cited all relevant tools in result section.
- “Determination of MIC” -> Which media was used?
: Thank you for your insightful comment. We mentioned the media information used in determining MIC.
Reviewer 2 Report
Comments and Suggestions for Authors
This manuscript presents an outstanding investigation into the dual antimicrobial and anticancer activities of marine-derived mitochondrial targeting sequences (MTSs). The work demonstrates rigorous experimental design, from bioinformatic screening of 3,923 mitochondrial proteins to comprehensive in vitro and in vivo validation of selected peptides. The discovery of MCCC1-MTS as a highly stable, low-toxicity therapeutic candidate with selective cancer cell targeting represents a significant advancement in the field of peptide-based therapeutics. While the study merits publication in International Journal of Molecular Sciences with minor revisions, several points require clarification:
1. The differential cytotoxicity between MCCC1-MTS and DBT-MTS warrants deeper analysis:
DBT-MTS's unexpected toxicity (60% HaCaT cell death at 150 μg/ml) contradicts the presumed mitochondrial specificity of MTSs. Please provide an appropriate explanation, such as what are the special characteristics of its amino acid composition/charge distribution?
How do the structural differences between MCCC1-MTS and DBT-MTS explain the differences in activity
2. Why were mitochondrial proteins from these 12 marine organisms selected for bioinformatics screening? What are the screening criteria? Is it possible that the screening criteria may overlook other potential species?
3. Please ensure strict adherence to IJMS style guidelines: Journal names should be consistently abbreviated;Author lists must be complete;Page ranges should use en-dashes;Article titles should not be italicized;etc.
- Figure 1B: Consider adding a table summarizing all 25 MTSs' antimicrobial spectra
This work fundamentally advances our understanding of MTSs as multifunctional therapeutic agents. The authors are commended for their rigorous approach and clinically relevant findings. With these revisions, the manuscript will reach its full potential as a significant contribution to the field.
Author Response
This manuscript presents an outstanding investigation into the dual antimicrobial and anticancer activities of marine-derived mitochondrial targeting sequences (MTSs). The work demonstrates rigorous experimental design, from bioinformatic screening of 3,923 mitochondrial proteins to comprehensive in vitro and in vivo validation of selected peptides. The discovery of MCCC1-MTS as a highly stable, low-toxicity therapeutic candidate with selective cancer cell targeting represents a significant advancement in the field of peptide-based therapeutics. While the study merits publication in International Journal of Molecular Sciences with minor revisions, several points require clarification:
1. The differential cytotoxicity between MCCC1-MTS and DBT-MTS warrants deeper analysis:
DBT-MTS's unexpected toxicity (60% HaCaT cell death at 150 μg/ml) contradicts the presumed mitochondrial specificity of MTSs. Please provide an appropriate explanation, such as what are the special characteristics of its amino acid composition/charge distribution?
How do the structural differences between MCCC1-MTS and DBT-MTS explain the differences in activity
: Thank you for your insightful comment. We revised the Discussion section to address the differential cytotoxicity between DBT-MTS and MCCC1-MTS. Specifically, we now describe that DBT-MTS exhibits a high proportion of hydrophobic amino acids (50%) and contains lysine residue, which may collectively contribute to its increased interaction with normal cell membranes and off-target cytotoxicity.
2. Why were mitochondrial proteins from these 12 marine organisms selected for bioinformatics screening? What are the screening criteria? Is it possible that the screening criteria may overlook other potential species?
: Thank you for your insightful comment. The 12 marine organisms used in our bioinformatics screening were selected based on their natural habitat in coastal waters near Korea and their traditional or potential use as food ingredients in Korea. This selection was made to prioritize species that are both readily accessible and generally recognized as safe (GRAS) for future application in functional food or therapeutic development.
The primary screening criteria included: (1) availability of annotated mitochondrial protein sequences in public databases (e.g., NCBI, UniProt); (2) ecological and geographical relevance to Korea; and (3) existing or potential use as food materials. We acknowledge that this selection approach may limit the taxonomic diversity and potentially overlook other marine species with bioactive MTSs. We plan to expand our screening in future studies to include a broader range of species from different marine environments to uncover additional candidates with therapeutic potential.
3. Please ensure strict adherence to IJMS style guidelines: Journal names should be consistently abbreviated;Author lists must be complete;Page ranges should use en-dashes;Article titles should not be italicized;etc.
: Thank you for your comment. We have carefully revised the reference list to strictly follow the IJMS formatting guidelines. Journal names have been consistently abbreviated, full author lists are provided, en-dashes have been used for page ranges, and article titles are presented in plain text without italics.
- Figure 1B: Consider adding a table summarizing all 25 MTSs' antimicrobial spectra
: Thank you for your insightful comment. We added the table summarizing all 25 MTS’s antimicrobial spectra as Figure S3 in supplementary materials
This work fundamentally advances our understanding of MTSs as multifunctional therapeutic agents. The authors are commended for their rigorous approach and clinically relevant findings. With these revisions, the manuscript will reach its full potential as a significant contribution to the field.